# Spatial Proximity of Immune Cell Pairs to Cancer Cells in the Tumor Microenvironment as Biomarkers for Patient Stratification

**DOI:** 10.3390/cancers17142335

**Published:** 2025-07-14

**Authors:** Jian-Rong Li, Xingxin Pan, Yupei Lin, Yanding Zhao, Yanhong Liu, Yong Li, Christopher I. Amos, Chao Cheng

**Affiliations:** 1Institute for Clinical and Translational Research, Baylor College of Medicine, Houston, TX 77030, USA; 233549@bcm.edu (J.-R.L.); xingxin.pan@bcm.edu (X.P.); yupei.lin@bcm.edu (Y.L.); yl10@bcm.edu (Y.L.); 2Section of Epidemiology and Population Sciences, Department of Medicine, Baylor College of Medicine, Houston, TX 77030, USA; yong.li@bcm.edu; 3Dan L Duncan Comprehensive Cancer Center, Baylor College of Medicine, Houston, TX 77030, USA; 4Department of Genetics, Stanford University School of Medicine, Stanford, CA 94305, USA; ydzhao@stanford.edu; 5Department of Internal Medicine, Division of Epidemiology, Biostatistics, and Preventive Medicine, University of New Mexico Comprehensive Cancer Center, Albuquerque, NM 87131, USA; ciamos@salud.unm.edu

**Keywords:** imaging mass cytometry, tumor microenvironment, prognosis, infiltrating immune cells, lung adenocarcinoma, triple-negative breast cancer

## Abstract

The tumor microenvironment (TME) is known to influence both disease progression and treatment outcomes in cancer. In this study, we analyzed high-dimensional imaging mass cytometry data from lung adenocarcinoma and triple-negative breast cancer to examine how the spatial positioning of immune cells relates to clinical outcomes. We developed a relative distance (RD) score that quantifies the proximity of immune cell pairs to cancer cells. This metric showed stronger associations with patient survival and treatment response than conventional features such as cell fraction or density. Our results suggest that spatial immune features may serve as robust biomarkers for patient stratification and personalized therapy.

## 1. Introduction

Cancer is a heterogeneous disease, with patients, even those with the same cancer type, exhibiting significant variability in prognosis [1,2]. Over the past decade, advancements in novel therapeutic strategies, particularly immune checkpoint blockade therapies (ICBTs), have significantly improved outcomes for many cancer types [3,4]. However, patient responses to these treatments vary dramatically—some experience long-term benefits, while others fail to respond or suffer severe adverse effects. To enhance clinical outcomes, there is an urgent need to develop prognostic and predictive biomarkers for patient stratification and personalized treatment guidance [5,6].

The tumor microenvironment (TME) plays a crucial role in cancer initiation, progression, metastasis, and therapeutic response [7,8]. Within the TME, immune cells can exert both anti- and pro-tumor effects. CD8+ cytotoxic T cells, B cells, and Natural Killer (NK) cells primarily contribute to anti-tumor immunity, whereas Regulatory T Cells (Tregs), M2 macrophages, and Myeloid-Derived Suppressor Cells (MDSCs) typically promote tumor progression. However, many infiltrating immune cells can have dual functions depending on their context. Pan-cancer analyses have underscored the strong prognostic impact of various immune cell populations, such as T cells and B cells, across multiple cancer types [9,10,11,12]. Furthermore, immune cell infiltration into the TME has been associated with patient response to therapy, particularly ICBTs [13,14,15,16,17,18]. For instance, tumor-infiltrating B cells and associated tertiary lymphoid structures (TLSs) have been identified as favorable biomarkers in breast, lung, sarcoma, melanoma, and other cancers, correlating with treatment sensitivity and extended survival following ICBT treatment [9,19,20]. While CD8+ T cells are generally regarded as anti-tumor effectors, their prognostic value remains controversial [21]. Some studies link CD8+ T cell infiltration to favorable prognosis and improved immunotherapy response [11,22,23], whereas others associate it with poor outcomes in cancers such as clear cell renal cell carcinoma (RCC), prostate cancer, and lymphoma [24,25,26,27].

Several factors influence the functional role and prognostic impact of immune cells in the TME, including cell subtype, activation state, spatial distribution, and interactions with other immune and tumor cells [28,29,30]. The relative infiltration levels of immune cells in tumors versus adjacent non-neoplastic tissues have been found to be more prognostic than their absolute infiltration levels in either tissue alone [31]. Moreover, studies have shown that immune cell ratios within the TME are more predictive of clinical outcomes than absolute cell counts [32,33,34]. For example, in ovarian cancer, a high CD8+ T cell to Treg ratio is linked to a favorable prognosis [33], whereas in lung squamous cell carcinoma, a high CD4/CD8 ratio is associated with poorer outcomes [32]. Computational analyses suggest that relative immune cell abundances provide deeper insights into their anti- or pro-tumor roles within the TME [35]. Additionally, the prognostic impact of major lymphocytes, including B cells, cytotoxic T cells, and T helper cells, is influenced by their spatial proximity to other immune cells, such as alternative macrophages [36].

Recently, imaging mass cytometry (IMC) has emerged as a powerful tool for characterizing the TME at single-cell resolution [37,38]. By measuring the expression levels of a panel of protein markers, IMC enables the simultaneous spatial mapping of diverse immune and tumor cell types. In this study, we leveraged large-scale IMC data to test the hypothesis that the relative spatial proximity of specific immune cell pairs to cancer cells is associated with clinical outcomes in cancer patients. Our analysis identified key immune cell pairs whose relative distances to cancer cells significantly correlate with the overall survival of patients with lung adenocarcinoma (LUAD). Additionally, we identified immune cell pairs that are highly predictive of patient response to immunochemotherapy or chemotherapy in triple-negative breast cancer (TNBC). Compared to conventional metrics such as cell density or cell fractions, relative distances to cancer cells offer a more effective set of biomarkers for prognostic stratification and treatment outcome prediction.

## 2. Methods and Materials

### 2.1. The IMC Datasets Collection

This study used IMC and matched clinical data from two previously published cohorts: LUAD and TNBC. The LUAD dataset was published by Sorin et al. [39] and obtained from Zenodo (DOI: 10.5281/zenodo.7760826). The TNBC dataset was published by Wang et al. [40] and accessed via Zenodo (DOI: 10.5281/zenodo.7990870).

#### 2.1.1. The LUAD Data

This dataset includes 416 IMC images, each corresponding to a single tumor sample from a patient with lung adenocarcinoma. Using a panel of 35 protein markers, cells were classified into 16 distinct types, including cancer cells, endothelial cells, and 14 immune cell types: CD163+ macrophages (Alt Mac), CD163- macrophages (CI Mac), CD8+ T cells (Tc), CD4+ helper T cells (Th), regulatory T cells (Treg), other T cells (Tother), classical monocytes (Cl Mo), non-classical monocytes (non-Cl Mo), intermediate monocytes (Int Mo), NK cells, dendritic cells (DC), mast cells, neutrophils, and undefined immune cells (Undefined). All LUAD samples were treatment-naïve primary surgical resections. The dataset also includes comprehensive clinical information, such as age group, sex, body mass index, smoking status, stage, histological subtypes, and overall survival data.

#### 2.1.2. The TNBC Data

This dataset comprises 1855 IMC images from 660 triple-negative breast cancer (TNBC) samples collected from 279 patients enrolled in a randomized clinical trial. Patients were randomly assigned to one of two treatment arms: (1) chemotherapy with neoadjuvant carboplatin and nab-paclitaxel alone or (2) immunochemotherapy combining the same chemotherapy regimen with atezolizumab, a PD-L1-targeting therapy. Patients were classified post-treatment as responders (pathological Complete Response, pCR) or non-responders (Residual Disease, RD). The chemotherapy arm included 62 responders and 79 non-responders, while the immunochemotherapy arm comprised 67 responders and 71 non-responders. Tumor samples were collected at three time points: baseline (*n* = 243), on-treatment (*n* = 207), and post-treatment (*n* = 210). For each sample, one or multiple IMC images were generated using an antibody panel targeting 43 protein markers. Based on these markers, cells were classified into 17 cancer cell types and 20 tumor microenvironment (TME) cell types, including endothelial cells, fibroblasts, myofibroblasts, PDPN+ stromal cells, CA9+ cells, and 15 immune cell types.

### 2.2. Calculation of Relative Distance of Non-Cancer Cell Pairs to Cancer Cells

We used the following procedure to calculate the relative distance of a pair of non-cancer cell types (X and Y) to cancer cells in an IMC image. First, for each cancer cell k, we calculated its distance to the nearest X cell. If no X cell was identified in the image, or if the closest distance exceeded 500 μm, we set the distance to 500 μm. This threshold exceeds typical cytokine diffusion ranges and provides a conservative upper bound for distal immune interactions, while ensuring spatial coverage within the IMC field of view.

Similarly, the distance to the nearest Y cell for each cancer cell was calculated. Second, the closest distances to X and Y cells for all cancer cells were averaged to obtain $\bar{d}x$ and $\bar{d}y$, respectively. Finally, the relative distance of X and Y to cancer cells in this image was calculated as$$RD\left(X\to Y\right)=\bar{d}x\;/\;(\bar{d}x+\bar{d}y)$$

A higher RD score for *X*→*Y* indicates that cancer cells tend to be farther from X cells than Y cells, meaning that cancer cells are less accessible to X cells compared to Y cells. For comparison purposes, we used the ratio density(X)/(density(X) + density(Y)) to represent the relative density of two cell types (X versus Y).

### 2.3. Normalization of RD-Scores

The RD score for a cell pair X→Y is largely influenced by the densities of X and Y cells. To adjust for this effect, we calculated normalized RD-scores (NRD-scores) using permutations. In each permutation, we shuffled the positions of all non-cancer cells in an IMC image and recalculated the RD scores for all non-cancer cell pairs. Based on 1000 permutations, we obtained null distributions of RD-scores, which approximated normal distributions. The original RD-scores were then normalized as follows:$$NRD=\left(RD-mean\left(RD^{\prime }\right)\right)\;/\;sd(RD\prime )$$
where *mean*(*RD*’) and *sd*(*RD*’) are the mean and standard deviation of the null distribution from permuted RD-scores (*RD’*). After adjusting for cell densities, the NRD-scores reflect the accessibility of X and Y cells to cancer cells via molecular interactions, such as cytokine-mediated effects.

### 2.4. Association of Features with Patient Prognosis

The LUAD IMC dataset provides overall survival information for all patients. Survival analyses were conducted to evaluate the association of RD-scores, cell fractions, densities, and other features with patient prognosis.

Univariable Cox regression models were used to determine the prognostic significance of individual features. Multivariable Cox regression models were adjusted for clinical factors such as age, sex, BMI, smoking status, stage, and histological subtype. Stratified analyses were performed within patient subgroups, and Kaplan–Meier plots were used to visualize survival differences. The R package “survival”, version 3.8.3, was used for these analyses.

### 2.5. Association of Features with Patient Response to Immunochemotherapy

The TNBC IMC data provides the treatment outcome (response or non-response) of patients treated by immunochemotherapy or chemotherapy. Using features defined based on the baseline IMC images, we assessed their classification accuracy to distinguish responders versus non-responders using Area Under the Curve (AUC) scores from Receiver Operating Characteristic (ROC) analysis. To quantify the prediction accuracy, we calculated their AUC (Area under the ROC curve) scores in classifying responders versus non-responders. AUC scores range from 0 to 1, where 0.5 indicates random prediction, and values above or below 0.5 indicate positive or negative correlations with treatment response, respectively. We computed rescaled AUC scores (AUC’ = 0.5 + |AUC − 0.5|) to quantify classification performance. The R package “ROCR”, version 1.0.11, was used for these analyses.

### 2.6. Statistical Analysis

Multivariable linear regression analysis was conducted to examine the association between RD scores and clinical factors in the LUAD IMC data: RD-score ~ Age + Sex + BMI + Smoking + Stage + HistologicalType

The estimated coefficients and *p*-values indicate the effect size and statistical significance of each factor. The Student’s *t*-test and Wilcoxon Rank Sum test were used for continuous variable comparisons, while Fisher’s exact test examined categorical associations. All statistical analyses were conducted in the R platform, version 4.3.1.

## 3. Results

### 3.1. Determinants of Immune Spatial Variability in Lung Adenocarcinoma

We used relative distance scores (RD-scores) to quantify the differential proximity of two distinct cell types to cancer cells in the tumor microenvironment (TME) based on cell positions and identities captured by IMC data (Figure 1A). A higher RD-score for a given cell pair X→Y indicates that X cells are farther from cancer cells compared to Y cells within an image. By definition, the RD-score for X→Y equals one minus the RD-score for Y→X, so we only considered one direction to avoid redundancy. The LUAD IMC dataset includes cancer cells and 15 non-cancer cell types, comprising endothelial cells and 14 immune cell types [39]. We calculated RD-scores for all possible 105 non-cancer cell pairs across all images. Hierarchical clustering analysis revealed variation in RD-scores across different samples. Some cell pairs consistently exhibited high (Cluster I) or low (Cluster III) RD-scores, while over 40% of cell pairs (Cluster II) displayed high variability (Figure 1B).

To determine the association between RD-scores and key clinical factors, we performed multivariable linear regression analysis across all cell pairs. For example, RD-scores for Tc→Th were significantly associated with smoking status, with lower values observed in smokers, suggesting that Tc cells are closer to cancer cells than Th cells in smokers (Figure 1C). Additionally, the solid histological subtype exhibited significantly lower RD-scores for Tc→Th compared to the lepidic subtype (Figure 1C). Figure 1D displays the cell pairs whose RD-scores were significantly associated with at least one clinical factor. Our analysis highlights that smoking status and histological subtypes substantially impact RD-scores for multiple cell pairs, whereas age, sex, BMI (Body Mass Index), and tumor stage have comparatively weaker associations (Figure 1D).

Most associations identified by multivariable linear regression models were validated through direct comparisons of RD-scores between relevant clinical groups. For instance, RD-scores for Th→Treg were significantly higher in smokers than in non-smokers (Figure 1E). Similarly, RD-scores for DC→Tc were significantly higher in early-stage tumors compared to late-stage tumor samples (Figure 1F). Additionally, significantly lower RD-scores for IntMo→Th were observed in the solid histological subtype compared to other histological types (Figure 1G). Collectively, these analyses demonstrate that the relative distances between different immune cell pairs and cancer cells are influenced by multiple clinical factors.

### 3.2. Association of RD-Scores with Patient Prognosis in Lung Adenocarcinoma

The RD-score for a cell pair reflects its differential capacity to reach and interact with cancer cells. Some cell pairs may play antagonistic roles in regulating cell growth and survival, leading us to hypothesize that RD-scores for certain cell pairs might be more strongly correlated with patient clinical outcomes than individual cell types alone.

To test this hypothesis, we systematically examined the association between RD-scores and overall survival (OS) in the LUAD IMC dataset. Univariable Cox regression analysis identified 25 significant cell pairs out of 105 at a significance level of 0.05 (False Discovery Rate, FDR < 0.05), including 14 protective (Hazard Ratio, HR < 1) and 14 hazardous pairs (HR > 1) (Figure 2A and Appendix A). At a stricter significance level (*p* < 0.01, unadjusted), 28 cell pairs were significantly associated with patient OS.

For comparison, we assessed the prognostic association of individual non-cancer cell types. While the fractions of B cells, T cells, and Int Mo cells were significantly associated with patient OS (*p* < 0.01), these associations were weaker compared to those observed with RD-scores of cell pairs (Figure 2B). When considering cell density instead of fraction, none of these cell types showed significant associations with patient OS (Figure 2C). Additionally, we measured the proximity of each cell type (X) to cancer cells by averaging the distances of all cancer cells to the nearest X cells. Only Int Mo cells showed a weak correlation with patient OS (Figure 2D).

In Figure 2E, we summarize the prognostic associations of all examined features, highlighting that analyzing the relative distances of different cell types to cancer cells can potentially yield more effective prognostic biomarkers. In Figure 2F, we illustrate the 25 significant cell pairs in a directed network, depicting interactions between prognostically favorable and unfavorable cell types. As shown, B cells, mast cells, and Th cells exhibited the highest out-degree, suggesting predominant anti-tumor roles in the TME. In contrast, Int Mo and Alt Mac cells had the largest in-degree, supporting their overall pro-tumor functions.

### 3.3. The Prognostic Association of RD-Scores for B→IntMo

Out of all significant cell pairs, the RD-score for B→IntMo exhibited the strongest prognostic association with an HR > 1, suggesting that patients whose cancer cells are more distant from B cells relative to Int Mo cells tend to have shorter survival times (Figure 3A). Consistently, a higher fraction and higher density of B cells, as well as a more distant distribution of B cells from cancer cells, were all associated with a favorable prognosis. In contrast, a higher fraction and higher density of Int Mo cells, along with a more distant distribution of Int Mo cells from cancer cells, were associated with an unfavorable prognosis (Figure 3A).

The distribution of RD-scores for B→IntMo is illustrated in Figure 3B. A larger proportion of patients (images) had RD-scores below 0.5, indicating that cancer cells were generally closer to B cells than to Int Mo cells. When patients were dichotomized into two groups using the median RD-score as the cut-off, those in the higher RD-score group exhibited significantly shorter overall survival (OS) compared to those in the lower RD-score group (Figure 3C). Furthermore, the association of RD-scores with OS remained significant after adjusting for major clinical variables, including age, sex, BMI, and tumor stage (Figure 3D). These findings suggest that the RD-score for B→IntMo serves as a robust biomarker for prognostic stratification of LUAD patients.

### 3.4. Normalized RD-Scores for Prognostic Analysis

In general, non-cancer cell types with high density in IMC images are expected to have shorter distances to cancer cells. Therefore, the relative distance between two cell types and cancer cells is largely influenced by their densities. To quantify this effect, we calculated the density ratio for all non-cancer cell pairs in the LUAD IMC data and correlated them with their RD-ratios across all samples. As expected, for most cell pairs, these values were highly correlated (Figure 4A), indicating that density differences predominantly determine RD-ratios. For example, RD-scores for B→AltMAC were strongly negatively correlated with the density ratio of these two cell types (R = −0.888), explaining approximately 80% of the variation in RD-scores (Figure 4B). Prognostic analysis identified 23 cell pairs significantly associated with patient OS based on their density ratio at *p* < 0.01 (10 protective and 13 hazardous, Figure 4C and Appendix A). However, the effect size and statistical significance were lower than those observed for RD-scores (Figure 2A).

Although cell density is a major factor, other variables, such as cytokine-mediated interactions and ligand-receptor signaling, can also influence the relative distances of cell type pairs to cancer cells. To account for these molecular interactions, we performed permutations by shuffling cell type labels in an image while preserving overall density, generating null distributions for RD-scores in all cell pairs. By normalizing RD-scores against the null distribution, we obtained normalized RD-scores (NRD-scores) that eliminated the contribution of cell density. Interestingly, many NRD-scores were significantly associated with patient prognosis (Figure 4D, Appendix A). For instance, when patients were stratified into two groups based on the NRD-scores for NK→Tc, the high-score group exhibited significantly shorter OS compared to the low-score group (Figure 4E). Furthermore, the significant cell pairs identified using NRD-scores differed substantially from those identified using raw RD-scores (Appendix A), suggesting that different cellular mechanisms contribute to their spatial positioning and interactions in the TME.

### 3.5. Association of RD-Scores with Treatment Response in Triple-Negative Breast Cancer

After assessing the prognostic impact of RD-scores, we examined the TNBC IMC dataset to determine their ability to distinguish responders from non-responders to immunochemotherapy (C&I) and chemotherapy (C). The dataset includes 17 cancer cell types and 20 TME cell types. We first assessed classification accuracy using conventional cell abundance features, including the fractions and densities of all cancer and TME cell subtypes (Figure 5A). The best classification accuracy was achieved by MHCI&IIhi density (a cancer cell type) in immunochemotherapy (AUC = 0.694) and PD-L1 + APCs fraction and density (a TME cell type) in chemotherapy (AUC = 0.644).

To evaluate whether relative distances to cancer cells could enhance classification accuracy, we calculated RD-scores for each of the 190 TME cell type pairs relative to all cancer cells (aggregated across all cancer types) and to each of the 17 specific cancer cell types. The resultant RD-scores were used to classify responders versus non-responders, with results shown in Figure 5B. Notably, many TME cell pairs achieved high classification accuracy, with the highest AUC reaching 0.794 for immunochemotherapy.

Interestingly, when overall cancer cells were used as a reference, classification accuracy was limited, with only a few significant TME cell pairs achieving a high AUC (>0.75 or <0.25) (Figure 5C). However, when using PD-L1 + GZMB+ cancer cells as the reference, several TME cell pairs exhibited high AUC scores (Figure 5D). For example, Treg→Endothelial achieved an AUC of 794, with significantly lower RD-scores in responders than non-responders to immunochemotherapy (Figure 5E). Likewise, CD56 + NK→Treg interactions in Helios+ cancer cells (Figure 5F) demonstrated high classification accuracy, with significant differences between responders and non-responders.

A similar trend was observed in the chemotherapy arm. As shown, when overall cancer cells were used as a reference, none of the TME cell pairs achieved significant classification accuracy (Figure 5G). However, when specific cancer cell types were used, several showed high classification accuracy. For example, when TCF1+ cancer cells were used as reference, multiple TME cell pairs yielded high AUC scores (Figure 5H). Notably, the CD4 + PD1 + T→Endothelial had a favorable AUC of 0.740, with significantly higher values in non-responders compared to responders to chemotherapy (Figure 5I). Similarly, the RD scores for CD20 + B→CD4 + TCF1 + T interactions with MHCI&IIhi cancer cells (Figure 5J) also exhibited significant differences between responders and non-responders.

## 4. Discussion

In this study, we explored the hypothesis that the differential proximity of specific immune cell pairs to cancer cells in the tumor microenvironment (TME) reflects their pro- or anti-tumor functions. Through a systematic analysis of non-cancer cell pairs associated with patient prognosis in LUAD, we demonstrated that the relative distances between certain immune cell pairs and cancer cells are more prognostic than traditional immunological metrics, including cell fractions, densities, and the distances of individual cell types to cancer cells. This relative distance framework compares immune cell types based on their spatial positioning around cancer cells, rather than viewing them in isolation. It better captures competitive or exclusionary immune dynamics that may help explain the clinical relevance of immune cell interactions. Notably, the RD-score for B→IntMo exhibited the strongest prognostic association (HR > 1), suggesting that increased proximity of B cells to cancer cells relative to IntMo cells correlates with improved prognosis.

B cells contribute to anti-tumor immunity by producing antibodies, promoting T cell activation, and releasing cytokines that enhance immune responses [17,18,41]. Consistently, our analysis revealed that a higher fraction of B cells was associated with prolonged patient overall survival (OS) in the LUAD IMC dataset. Human monocytes are classified into three major subsets: classical (CD14+CD16−), non-classical (CD14dimCD16+), and intermediate (CD14+CD16+). While classical and non-classical monocytes showed no correlation with patient survival, the fraction of intermediate monocytes was negatively associated with OS. Although the role of intermediate monocytes in the TME remains poorly understood, the strong prognostic association of the RD-score for B→IntMo suggests opposing functions between these two immune cell types, warranting further investigation through controlled experimental studies.

Cell proximity in the TME is determined by two primary factors: (i) cell densities and (ii) interactions mediated by cytokines and receptor-ligand signaling. While cell–cell interactions can be either cooperative or antagonistic, their influence can be assessed by comparing observed distances with expected distances under random distributions. Our findings indicate that RD-scores for most cell pairs to cancer cells are largely dictated by density differences. If the density of X cells is higher than that of Y cells, X cells are more likely to be closer to cancer cells, leading to an RD-score for X→Y greater than 0.5. To account for these density-driven effects, we performed permutations that shuffled cell type labels while preserving overall densities. This allowed us to generate null distributions of RD-scores for each cell pair and derive normalized RD-scores (NRD-scores) that mitigate density-related biases. Interestingly, while NRD-scores for some cell pairs were significantly associated with patient prognosis, their patterns of association differed substantially from those of RD-scores. These results suggest that while immune cell infiltration levels shape their spatial distribution within the TME, additional factors such as cytokine signaling and cell–cell interactions also modulate proximity to cancer cells.

Our findings further demonstrate that the relative distances of non-cancer cell pairs to cancer cells can enhance predictions of patient response to immunochemotherapy or chemotherapy, as evidenced by the TNBC IMC dataset. The RD-score for Endothelial→Treg relative to PD-L1 + GZMB+ cancer cells achieved an AUC of 0.794, significantly outperforming the highest AUC of 0.691 obtained using conventional immunological features. This underscores the potential of RD-scores as superior prognostic and predictive biomarkers compared to conventional measures such as cell fraction or density.

Several limitations should be considered when interpreting our findings. The LUAD dataset lacked mutational profiling such as *EGFR*, *ALK*, or *KRAS* status, which are key determinants of prognosis in lung cancer and could influence spatial associations with survival. The TNBC dataset did not include molecular subtype annotations (e.g., immunomodulatory, luminal androgen receptor), limiting our ability to explore subtype-specific spatial patterns. Our analysis was also based on fixed tissue samples, which provide a single static view of the tumor microenvironment; each IMC image represents one time point, and fixation may introduce spatial artifacts or snapshot bias. Moreover, spatial heterogeneity within tumors presents challenges for reproducibility. In the TNBC cohort, we observed notable differences in immune composition and RD-scores across multiple IMC images from the same patient. Although we computed RD-scores per image rather than averaging across regions, such variation suggests that tumor sampling location can significantly affect spatial metrics, particularly in patients with only a single-region sample. This highlights the importance of multiregional sampling in capturing the complexity of the tumor microenvironment, as demonstrated in our previous transcriptomic study of NSCLC [42]. Future studies that integrate genomic data, subtype stratification, multiregional sampling, and longitudinal spatial profiling are warranted to address these limitations and validate the clinical utility of spatial immune metrics.

LUAD and TNBC were selected for this study based on the availability of large-scale, high-quality IMC datasets with corresponding clinical annotations. While these cancers differ in biological characteristics, they provided distinct yet complementary settings to test the RD-score framework—overall survival in LUAD and treatment response in TNBC. Although we did not identify immune cell pairs with consistent associations across both cancers, our results illustrate the broader relevance of spatial immune organization as a predictor of clinical outcomes in diverse tumor types.

## 5. Conclusions

In summary, our study highlights the utility of RD-scores in capturing the functional roles of immune cells in the TME. The relative distances of specific immune cell pairs to cancer cells reflect their pro- or anti-tumor functions and serve as robust prognostic and predictive biomarkers. This framework provides a novel strategy for patient stratification and personalized treatment selection, paving the way for more precise immunotherapy applications.

## Figures and Tables

**Figure 1 cancers-17-02335-f001:**
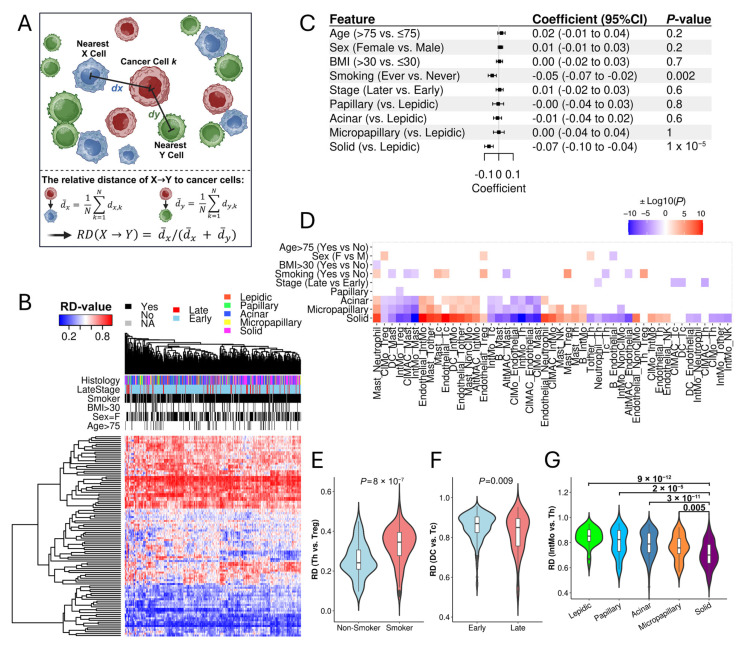
Association of RD-Scores with Clinical Factors in LUAD. (**A**) Schematic diagram to show the calculation of RD-score to quantify the relative proximity of immune cell pairs (X, Y) to cancer cells. Higher RD-scores indicate X cells are farther from cancer cells than Y cells. (**B**) Hierarchical clustering of LUAD samples based on the RD-scores for 105 non-cancer cell pairs. Rows represent immune cell pairs, and columns represent tumor samples. (**C**) The association of RD-scores for Tc→Th with clinical variables determined by multivariable linear regression analysis. (**D**) The association of RD-scores with clinical factors. The color scale represents log10-transformed *p*-values, with red indicating positive and blue indicating negative associations. (**E**) The RD-scores for Th→Treg are significantly higher in smokers than non-smokers. (**F**) The RD-scores for DC→Tc are significantly higher in the early stage than late-stage samples. (**G**) The RD-scores for IntMo→Th vary significantly across histological subtypes.

**Figure 2 cancers-17-02335-f002:**
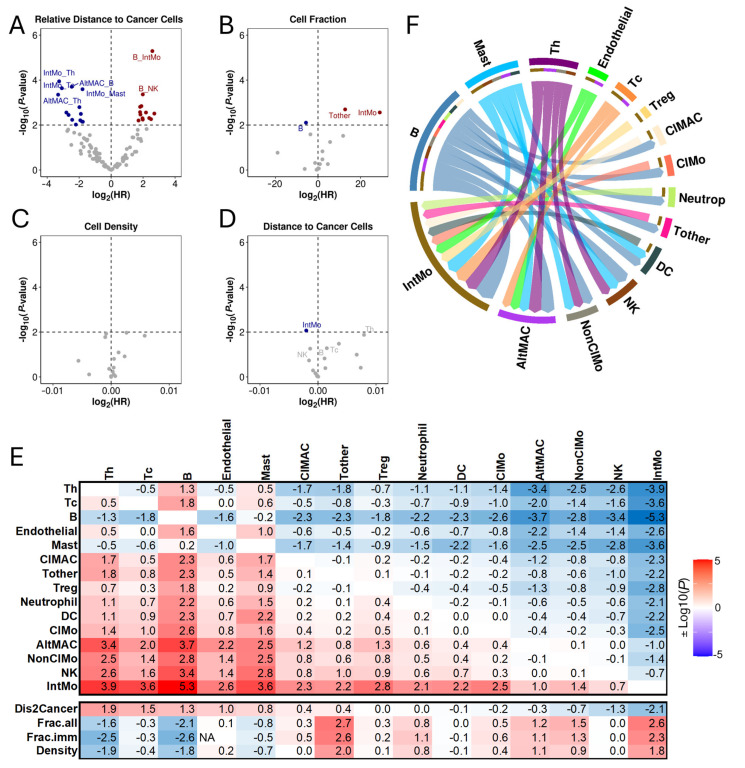
Association of RD-scores with patient overall survival in LUAD. (**A**) Volcano plots displaying the association of the RD-scores for all non-cancer cell pairs with patient prognosis. (**B**–**D**) Association of cell fraction, cell density, and distance to cancer cells with patient prognosis. (**E**) Heatmap showing the correlation between different immunological features (RD-scores, cell fractions, cell densities) and patient prognosis. The color scale represents log10-transformed *p*-values, with red indicating positive and blue indicating negative associations. (**F**) Circos plot illustrating significant interactions among different immune cell types in the TME. Each interaction represents a cell pair with RD-scores significantly associated with prognosis (FDR < 0.05). Arrows indicate directionality, originating from the favorable cell type and pointing toward the unfavorable cell type.

**Figure 3 cancers-17-02335-f003:**
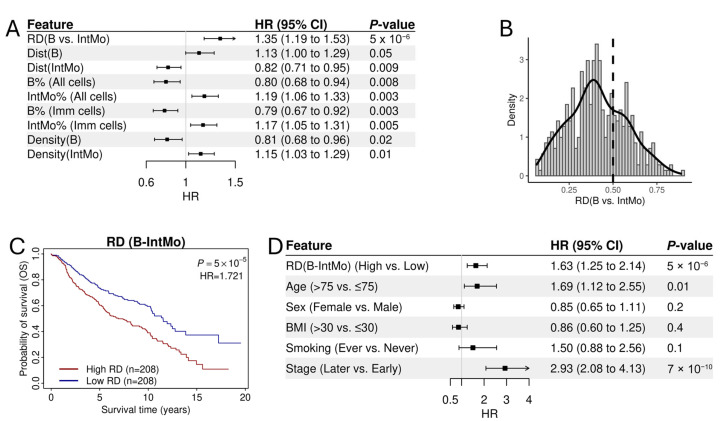
Prognostic association of the RD-scores for B→IntMo in LUAD. (**A**) The RD-scores for B→IntMo are more prognostic than conventional immunological features. (**B**) Distribution of RD-scores for B→IntMo across all samples. (**C**) Kaplan–Meier survival analysis comparing overall survival between patients with high and low RD(B→IntMo) scores. (**D**) The RD-scores for B→IntMo remain an independent prognostic factor after adjusting for clinical factors, including age, sex, BMI, smoking history, and tumor stage.

**Figure 4 cancers-17-02335-f004:**
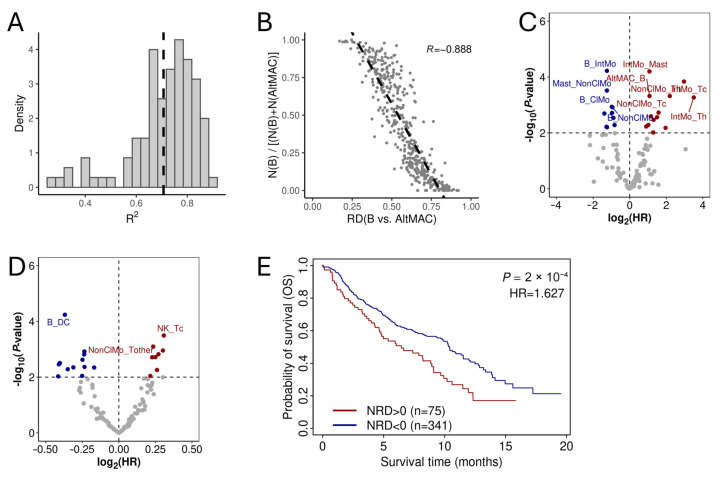
Prognostic association of NRD-scores (normalized RD-Scores). (**A**) Distribution of R^2^ values representing the correlation between RD-scores and cell density ratios across all samples for all non-cancer cell pairs. The vertical dashed line indicates the median R^2^ value. (**B**) Scatter plot showing the correlation between the RD-scores and the relative abundance of B→AltMAC. (**C**) Volcano plot displaying the prognostic association of the relative abundance for immune cell pairs with patient overall survival. (**D**) Volcano plot illustrates the prognostic significance of NRD-scores. (**E**) Kaplan–Meier survival curves comparing patients with high versus low NRD-scores for NK→Tc.

**Figure 5 cancers-17-02335-f005:**
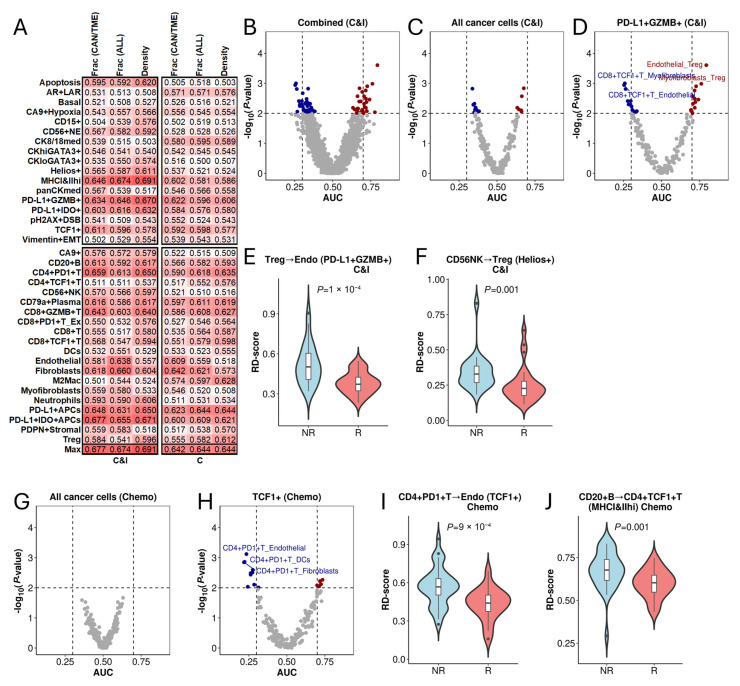
Association of RD-Scores with treatment response in TNBC. (**A**) Heatmap displaying the accuracy of conventional immunological features for classifying patient response to immunochemotherapy (C&I) and chemotherapy (C). The values are AUC scores with the bottom row indicating the maximum accuracy achieved by each feature. Darker red indicates a higher AUC value. (**B**–**D**) Volcano plots showing the association between RD-scores and patient response to immunochemotherapy. In (**B**), results using all cancer cells or each of the 17 cancer cell subtypes as the reference were combined. In (**C**), all cancer cells were used as the reference, and in (**D**), PD-L1 + GZMB+ cancer cells were used as the reference. (**E**) The RD-scores for Treg→Endothelial to PD-L1 + GZMB+ cancer cells are significantly lower in responders (R) than non-responders (NR). (**F**) The RD-scores for CD56 + NK→Treg to Helios+ cancer cells are significantly lower in responders than non-responders. (**G**) When all cancer cells are used as the reference, none of the TME cell pairs achieved significant association with patient response to chemotherapy. (**H**) Association of RD-scores with patient response to chemotherapy when TCF1+ cancer cells were used as reference. (**I**) The RD-scores for CD4 + PD1 + T→Endothelial to TCF1+ cancer cells are significantly lower in responders than non-responders to chemotherapy. (**J**) The RD-scores for CD20 + B→CD4 + TCF1 + T to MHCI&IIhi cancer cells are significantly lower in responders than non-responders to chemotherapy.

## Data Availability

The original TNBC IMC data used in this study was obtained from Wang et al. [40] and is publicly available on Zenodo (DOI: 10.5281/zenodo.7990870). The LUAD IMC data was obtained from Sorin et al. [39] and is available on Zenodo (DOI: 10.5281/zenodo.7760826). Processed datasets and analysis scripts generated in this study are available from the corresponding author upon reasonable request.

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
