# Peer review of "Spatial Proximity of Immune Cell Pairs to Cancer Cells in the Tumor Microenvironment as Biomarkers for Patient Stratification"

_cancers, 2025, doi:10.3390/cancers17142335_

Round 1
Reviewer 1 Report
Comments and Suggestions for Authors
In the following work, Jian-Rong Li et al. demonstrate that the relative spatial proximity of specific immune cell pairs to cancer cells is associated with clinical outcomes in cancer patients, using data obtained by Imaging Mass Cytometry (IMC).
The work presents a novel strategy to be able to personally characterize patients with lung cancer and triple-negative breast cancer.
Minor points
- In Fig. 1, the axes of the figures/diagrams are not clearly legible.
- In Fig. 2 E the axes of the figures cannot be clearly read.
Reviewer 2 Report
Comments and Suggestions for Authors
Indeed, the spatial aspect may play a crucial role in immune cell function, as both direct cell-cell contact (e.g., T and B cells or APCs and T cells) and proximity for paracrine signaling are important. In this regard, the study is novel, relevant, and will undoubtedly benefit the scientific community. However, several issues require clarification, and I request the authors to address them:
What is the effective diffusion distance of cytokines? Why was a 500 μm threshold chosen for analysis?
The Methods section indicates that the analysis focused on the relative proximity of immune cells to tumor cells (comparing two cell types) rather than measuring absolute distances. What is the rationale for this approach? What advantages does it offer over absolute distance quantification?
Were dynamic parameters assessed at multiple time points (e.g., three stages of disease progression or treatment)?
For lung cancer patients, mutational profiling (e.g., EGFR, ALK, KRAS) must be included, as it is a critical determinant of clinical outcomes. Without this information, the lung cancer findings cannot be properly interpreted.
How reproducible is this spatial analysis? For instance, were comparisons made between different tumor regions from the same patient?
Since fixed samples were used, the limitations of this approach (e.g., potential artifacts, snapshot bias) should be explicitly discussed. There is no certainty whether the same spatial patterns would hold at different time points for a given patient.
How did you address ethical considerations when working with patient-derived imaging data (e.g., de-identification, institutional review board approval)?
Reviewer 3 Report
Comments and Suggestions for Authors
Li et al. investigate whether the distance between a cancer cell and neighboring stromal cells in the microenvironment may be predictive of therapeutic and survival outcomes. Overall, the manuscript is well-written and discovers correlations between stromal cells and their proximity to cancer cells, particularly monocytes and B cells in lung cancers and T cells in triple negative breast cancers. While no causative or mechanistic studies are included, the data presents some interesting avenues to pursue for future studies. However, there are a few points to address:
- Important information regarding IMC data sources are buried in the data availability statement and acknowledgements. These should be well described in the methods section.
- The methods section needs to include a summary of the markers used to identify the cell populations and subpopulations. This will be especially important to understand the 17 cancer cell types mentioned in the TNBC datasets.
- Many of the figure subpanels are too small to read, including Figures 1B, highly slanted text in 1D, Axis titles in Figures 1E-F-G, 2E, and 5A. In addition, the volcano plots often have overlapping text that make it difficult to identify which populations are significant and the colors are too pale to easily see. Please use a darker red and blue for volcano plots and their text.
- The authors describe that significant correlations are primarily in the TNBC studies when different cancer cell subtypes are evaluated and limited when overall cancer cells are considered together. Additional evaluation to determine if there are correlations based on the known molecular subtypes of TNBC (e.g: immunomodulatory, luminal androgen receptor, etc.) may lead to more rigorous or significant findings.
- The Discussion section is under-developed. It should include a discussion on the limitations of the authors’ study. Furthermore, the combination of the two investigated cancers (lung and TNBC) do not have strong rationale for why both are presented. If there were commonalities found by the authors’ for the clinical relevance of specific cell combinations in both cancers, this should be discussed here. If there were no commonalities found regarding stromal cell proximity to cancer cells, then there should be a discussion of why they are so different.
Round 2
Reviewer 2 Report
Comments and Suggestions for Authors
The authors have responded to my questions; the article may be accepted.